

# 1 Reactive Organic Carbon Air Emissions from Mobile Sources in the
# 2 United States

Benjamin N. Murphy[1*], Darrell Sonntag[2], Karl M. Seltzer[3], Havala O. T. Pye[1], Christine Allen[4],
Evan Murray[5], Claudia Toro[5], Drew R. Gentner[6], Cheng Huang[7], Shantanu Jathar[8], Li Li[6],
Andrew A. May[9], and Allen L. Robinson[10]
[1]Center for Environmental Measurement and Modeling, US Environmental Protection Agency, Research Triangle
Park, North Carolina 27711, United States
[2]Department of Civil and Construction Engineering, Brigham Young University, Provo, Utah 84602, United States
[3]Office of Air quality Planning and Standards, US Environmental Protection Agency, Research Triangle Park, North
Carolina 27711, United States
[4]General Dynamics Information Technology, 79 T.W. Alexander Drive, Research Triangle Park, NC 27709, United
States
[5]Office of Transportation and Air Quality, US Environmental Protection Agency, Ann Arbor, Michigan 48105, United
States
[6]Department of Chemical and Environmental Engineering, Yale University, New Haven, CT 06511, United States
[7]State Environmental Protection Key Laboratory of Cause and Prevention of Urban Air Pollution Complex, Shanghai
Academy of Environmental Sciences, Shanghai, 200233, China
[8]Department of Mechanical Engineering, Colorado State University, Fort Collins, Colorado 80523, United States
[9]Department of Civil, Environmental and Geodetic Engineering, Ohio State University, Columbus, Ohio 43210,
United States
[10]Department of Mechanical Engineering, Carnegie Mellon University, Pittsburgh, Pennsylvania 15213, United
States; Carnegie Mellon University Africa, BP 6150 Kigali, Rwanda
*Correspondence to: Benjamin N. Murphy (murphy.ben@epa.gov)
**Abstract**: Mobile sources are responsible for a substantial controllable portion of the reactive organic carbon (ROC)
emitted to the atmosphere, especially in urban environments of the United States (U.S.). We update existing methods
for calculating mobile source organic particle and vapor emissions in the U.S. with over a decade of laboratory data
that parameterize the volatility and organic aerosol (OA) potential of emissions from onroad vehicles, nonroad
engines, aircraft, marine vessels, and locomotives. We find that existing emission factor information from teflon filters
combined with quartz filters collapses into simple relationships and can be used to reconstruct the complete volatility
distribution of ROC emissions. This new approach consists of source-specific filter artifact corrections and state-of-
the-science speciation including explicit intermediate volatility organic compounds (IVOCs), yielding the first
bottom-up volatility-resolved inventory of U.S. mobile source emissions. Using the Community Multiscale Air
Quality model, we estimate mobile sources account for 20-25% of the IVOC concentrations and 4.4-21.4% of ambient
OA. The updated emissions and air quality model reduce biases in predicting fine-particle organic carbon in winter,
spring, and autumn throughout the U.S. (4.3-11.3% reduction in normalized bias). We identify key uncertain
parameters that align with current state-of-the-art research measurement challenges.

## 37 1. Introduction

Ambient particulate matter (PM) and ozone ($O_3$) have detrimental impacts on human health and the environment (U.S.
EPA, 2019, 2020c; Pye et al., 2021) with disparate impacts across societal groups (Tessum et al., 2021). Non-methane
organic gases (NMOG) are precursors to PM and O3, and reducing NMOG could reduce criteria pollutants and their
associated mortality throughout the United States (U.S.) (Pye et al., 2022a). Mobile source emissions continue to be
a major contributor to modern anthropogenic NMOG emissions. In contrast to other NMOG sources such as
vegetation, mobile emissions have been reduced through successful regulatory policy and the introduction of cleaner
engine and control technologies (Lurmann et al., 2015; Gentner et al., 2017; Winkler et al., 2018; Bessagnet et al.,



2022). Yet, effective management of urban and regional air quality still depends on accurate and detailed
characterization of the carbon-containing compounds emitted by mobile sources.
Fossil-fuel combustion emissions comprise thousands of organic compounds with widely varying volatility,
depending on source type (Drozd et al., 2018; Lu et al., 2018). The lowest volatility compounds are emitted principally
in the particle phase and are typically classified as primary organic aerosol (POA). Conventionally this portion of
emissions is sampled using filters which are weighed or processed off-line with thermal-optical techniques, solvent
extraction, and other methodologies (Chow et al., 1993; Birch and Cary, 1996; U.S. EPA, 2022b). The highest
volatility NMOGs are emitted in the gas-phase and enhance $O_3$ formation when oxidized in the atmosphere, a process
that also enhances PM mass via secondary organic aerosol (SOA) formation. U. S. EPA emission tools like the MOtor
Vehicle Emission Simulator (MOVES) (U.S. EPA, 2020b) and the SPECIATE database (U.S. EPA, 2020a) provide
emission estimates and speciation for POA (assumed to be nonvolatile) and NMOGs. The 'Conventional' path in Fig.
1 depicts this process. However, laboratory and field measurement campaigns have demonstrated that much of the
mobile source POA is subject to gas-particle partitioning and filter sampling artifacts, semivolatile, which complicates
the interpretation of filter-based measurements (Robinson et al., 2010; Bessagnet et al., 2022). These compounds
principally include (Table 1) semivolatile organic compounds (SVOCs) and intermediate volatility organic
compounds (IVOCs), with IVOCs being key contributors to filter artifacts (May et al., 2013a, b). Accurately
representing SVOCs and IVOCs is important because they are SOA precursors and are underestimated in
contemporary models and emission databases (Gentner et al., 2012; Tkacik et al., 2012; Zhao et al., 2014; Zhao et al.,
2015, 2016b).
Some air quality models (AQMs) have incorporated semivolatile organic compounds (SVOCs) and IVOCs by
adapting emissions inputs either with a data pre-processing step or during the AQM runtime (Murphy and Pandis,
2009; Shrivastava et al., 2011; Ahmadov et al., 2012; Bergström et al., 2012; Koo et al., 2014; Woody et al., 2015;
Zhao et al., 2016a; Woody et al., 2016; Jathar et al., 2017b; Murphy et al., 2017). However, these approaches rely on
broad application of assumptions that may not be appropriate for specific source types since sampling artifacts will
bias low-emitting and high-emitting sources differently (Robinson et al., 2010). As emissions from individual
combustion sources are continually reduced in response to tightening regulations, accounting for these potential biases
becomes important. Bottom-up approaches are needed that revise emission factors and speciation profiles for
individual source types. Datasets like this exist for some areas like Europe (Manavi and Pandis, 2022), Japan (Morino
et al., 2022) and China (Chang et al., 2022).
This paper documents the transition of U. S. EPA mobile emission tools from the conventional paradigm that considers
operationally defined particulate organic matter (OM) and NMOG emission factors and speciation to one that
accommodates the full complexity of atmospheric carbon-containing trace pollutants. To accomplish this, we consider
total Reactive Organic Carbon (ROC), defined by Saffediene et al. (2017) and Heald and Kroll (2020) as all reactive
organic compound mass across gas and particle phases excluding methane. We catalogue updates to 51 diverse mobile
source categories across multiple categories and engine, fuel, and control types. Further, we demonstrate procedures
for integrating existing inventory emission factors with state-of-the-art chemical composition measurements, pointing



out where critical uncertainties could be further resolved in the future. Finally, we document the impact the updates
have on source-specific and sector-wide emissions as well as regional-scale pollutant formation and transport
predicted by an updated version (2020) of the Community Multiscale Air Quality (CMAQ) regional-scale AQM.
**2. Materials and Methods**
**2.1 Mobile Emission Modeling**
To develop the new framework and estimate potential impacts from speciation updates, we used existing estimates for
2016 annual mobile emissions for the contiguous U.S. We considered five categories including onroad, nonroad, air,
rail, and marine. The MOVES3 model predicts emissions for onroad and nonroad sources using county-level fleet
properties and activity data. The dominant U.S. onroad vehicle sources are light-duty gasoline cars and trucks and
heavy-duty diesel trucks. Nonroad emission sources include construction, agricultural, and lawn equipment as well as
nonroad recreational vehicles. The Aviation Environmental Design Tool (AEDT), maintained by the Federal Aviation
Administration, predicts landing, taxi, and take-off emissions for aircraft and emissions from ground support
equipment (Faa, 2022). Rail emissions are calculated using confidential line-haul activity data that were summarized
at the county-level, while rail-yard emissions are based on supply fuel use and yard switcher counts provided by
companies (U.S. EPA, 2022a). Marine emissions include both port and underway conditions for large, generally
international ships, vessels, and smaller boats operating near shore (U.S. EPA, 2022a). The MOVES3 model predicts
emissions from recreational boats as part of the nonroad recreational equipment category.
We also collected national total annual fuel usage data for each source from the models to calculate an effective fuel-
based OM emission factor (see section S1). These effective emission factors range from 1-20 mg (kg-fuel)$^{-1}$ for the
newest gasoline, diesel, and compressed natural gas (CNG) vehicles to over 6000 mg (kg-fuel)$^{-1}$ for nonroad gasoline
two-stroke engines. In the process of reviewing each mobile source OM emission rate, we discovered and corrected
several minor errors and limitations to compressed natural gas sources and uncontrolled nonroad diesel exhaust (see
section S2).
**2.2 Reactive Organic Carbon (ROC)**
To accurately simulate the behavior of mobile emissions, we must consider total ROC which includes organic carbon
(OC) and non-carbon mass from compounds from the most volatile species like ethane and formaldehyde to
chemically complex, high molecular weight compounds (e.g. oligomers) (Heald and Kroll, 2020). Conventional
metrics for reporting OM and NMOG are operationally defined based on measurement methods and conditions;
therefore, they are difficult to compare across tests and among other ROC sources. Furthermore, uncertainties are
introduced when they are speciated with profiles measured at different conditions. To improve standardization, we
introduce two new metrics: CROC (condensable reactive organic carbon) and GROC (gaseous reactive organic
carbon). CROC is defined as compounds with saturation concentration ($C^*$) less than 320 µg m$^{-3}$ (Table 1), with this
boundary corresponding to $n$-alkanes with 20±1 carbon atoms. CROC includes SVOCs ($0.32 < C^* \leq 320$ µg m$^{-3}$) and
low volatility organic compounds (LVOCs; $C^* \leq 0.32$ µg m$^{-3}$). Whereas, GROC is defined as the sum of compounds
with $C^*$ greater than 320 µg m$^{-3}$ corresponding to IVOCs ($320 < C^* \leq 3.2$ x $10^6$ µg m$^{-3}$) and volatile organic compounds
(VOCs; $C^* > 3.2$ x $10^6$ µg m$^{-3}$) (Donahue et al., 2009; Murphy et al., 2014). CROC and GROC align with well-known



categories in the volatility basis set (VBS) space, so they may be applied straight-forwardly to speciation profiles in
recent literature containing both explicit compounds and lumped groups.
We apply a two-step methodology to process gas- and particle-phase emissions ('ROC' path in Fig. 1). First, we
estimate total GROC and CROC emissions from existing NMOG and OM emission factors, respectively, while
considering measurement uncertainties like sampling setup losses (e.g. tubing) and filter artifacts. We then speciate
GROC and CROC using state-of-the-science profiles. For GROC, these include explicit IVOC compounds where
available and lumped IVOC groups distinguished by their saturation concentration and functionality. The
methodology for processing CROC emissions similarly uses volatility profiles from recent literature.

**2.2.1 GROC Emissions and Speciation**

Total NMOG emissions are measured from mobile emissions by combining total hydrocarbons (THC) with carbonyl
compounds and subtracting methane (see section S3) (Kishan et al., 2006; May et al., 2014). Lu et al. (2018) compiled
measurements for onroad vehicles, nonroad equipment, and an aircraft turbine engine. That study concluded that
methods using heated sampling and a heated flame-ionization detector (FID) can capture both IVOCs and VOCs, but
that speciation methods like canister or tedlar bag sampling analyzed with gas-chromatography-FID miss essentially
all IVOCs due to wall losses to the sampling materials. Assuming that NMOG emission rates are based on heated FID
sampling, we set GROC emission rates equal to total NMOG emission rates across all sources, and we speciated
GROC emissions using profiles that include VOCs and IVOCs.
Many studies have reported speciated organic gases normalized to total IVOC or VOC (Lu et al., 2018; Jathar et al.,
2017a; Zhao et al., 2015, 2016b; Huang et al., 2018; Drozd et al., 2018). A key parameter used to integrate these data
is the IVOC/NMOG ratio (see section S4), which ranges from ~4.6% for gasoline vehicle cold start exhaust to 67%
for marine residual oil. Gasoline fuel evaporation profiles of GROC were assumed to be the same as NMOG since
IVOCs are not expected to contribute substantially to those emissions (Gentner et al., 2012). The profile for whole
diesel fuel evaporation was updated to be consistent with fuel characterization in Gentner et al. (2012) (see Section
S1c). SPECIATEv5.1 contains thousands of explicit species and many mixtures of compounds (e.g. oils, unspeciated
terpenes, etc.) reported by previous studies. Recent studies have constrained the unknown portion of IVOCs and VOCs
with lumped groups resolved by volatility and often by structure/functionality features (e.g. branched, cyclic,
oxygenated, etc.). We leverage the representative compound structures in SPECIATE developed by Pye et al. (2022b)
to classify these emissions by functional groups, and their subsequent atmospheric chemistry. Table S2 summarizes
the new IVOC profiles. Species-based ozone and OA potential were calculated for each emission source using
relationships from Seltzer et al., (2021) which were expanded by Pye et al. (2022b)

**2.2.2 CROC Emissions and Speciation**

We estimate effective OM emission factors using the MOVES-predicted national total OM emissions normalized to
the total fuel usage for each source (see section S1). The MOVES model relies on conventional measurements of total
PM emissions sampled and weighed on Teflon filters. The SPECIATE database, meanwhile, stores the weight percent
of OC measured by thermal optical techniques from samples collected on quartz filters (U.S. EPA, 2022b) normalized
by coincident bulk PM measurements from the Teflon filter (see section S5). SPECIATE also applies a source-



dependent OM/OC factor to adjust for non-carbon organic mass (i.e. hydrogen, oxygen), which represents OM once
added to OC (Table S1a) (Reff et al., 2009; Simon et al., 2011). Previous studies have demonstrated that OM emission
factors vary with changing temperature and OM loading (Lipsky and Robinson, 2006; Robinson et al., 2010; May et
al., 2013b, a; Jathar et al., 2020). AQMs that take this behavior into account typically distribute OM emissions among
volatility bins using reference distributions. May et al. (2013b, a) constrained parameters for calculating volatility-
resolved emissions assuming OC is measured on a quartz filter. Although this approach performs well for average
cases, it is less accurate when applied to sources that are low or high emitting, for which absorptive partitioning biases
are more substantial (Fig. 2). For an exceedingly low-emitting source (low OM loading), SVOC emissions that would
normally partition to the particle phase under ambient conditions could go undetected as they pass through the filter.
Additionally, reported OM emissions are sometimes artifact-corrected using a secondary quartz filter behind the
Teflon filter sample, which allows for adsorbed SVOCs and IVOCs to be neglected. Because these corrections are not
uniformly applied across all studies, May et al. (2013b, a) reported reference volatility profiles assuming OM emission
factors had not been adsorptive-artifact corrected. Yet this is not always applicable for the emission rates informing
MOVES and must be resolved at the source level based on the underlying emission data. To address both adsorptive
and absorptive partitioning biases, we apply CROC/OM parameterizations developed from detailed measurement data
and informed by filter-based OM emission factors (see section S6) (May et al., 2013b, a; Huang et al., 2018; Jathar et
al., 2020). The method accounts for filter artifact corrections by adding missing SVOC emissions for low OM-loading
tests and neglecting IVOCs and higher-volatility SVOCs that would be captured on the front filter during high OM-
loading tests. The CROC/OM parameterization for onroad gasoline is based on data from 64 vehicles and so is more
robust than the parameterization for onroad heavy-duty diesel with particulate filters (DPF), which is based on 3
vehicles (Section S7), or the aircraft engine parameterization, which is based on one sample. More work is needed to
better constrain the CROC/OM parameters.
The impact of this new approach for translating inventory OM emissions is shown in Fig. 2. We use the onroad
gasoline light-duty cold start volatility profile in Table S5 to estimate the effective ambient organic aerosol emission
factor at 298 K and $C_{OA}$ equal to 10 μg m$^{-3}$ given a filter-based OM emission factor in mg kg$^{-1}$ fuel. Also shown are
trends using parameters reported by Robinson et al (2007) and Lu et al. (2020), which have been used in contemporary
air quality models. The filter-based OM emission factor ($EF_{OM}$) is multiplied by the volatility distribution, and VBS
partitioning theory (Eq. 1) is used to calculate the effective ambient OA emission factor ($EF_{OM,Amb}$):
$$EF_{OM,Amb} = EF_{OM} \sum_{i=1}^{n_{tot}} \frac{\alpha_i}{1 + C_i^*/10} \qquad (1)$$
where $n_{tot}$ is the number of volatility parameters in the vector α. The 'Lu et al.' and 'Robinson et al.' lines are directly
proportional to the nonvolatile emission factor because they do not consider nonlinear dependence on the filter-based
OM emission factor. Meanwhile, the ROC approach enhances emissions at low emission factors (to correct for SVOC
breakthrough) and reduces them at high emission factors (to remove IVOCs partitioning to the filter). Also shown on
Fig. 2 are filter-based OM emission factors for PreTier 2, Tier 2 (2001-2004), and Tier 2 (2004+) vehicles, which
exhibit emissions reductions with newer standards. For the older vehicles, the 'Lu et al.' and 'Robinson et al.'



approaches give similar estimates for effective ambient OM as the new approach, but as emission factors decrease,
those methods may overpredict evaporation and underpredict the particle emission factors. At the lowest OM emission
factors, even using the nonvolatile approach may underpredict effective ambient OA emission factors because
significant SVOCs could have broken through the filter and should be considered for ambient partitioning.
We did not adjust GROC emissions in response to CROC/OM conversion, but the sum of total ROC emissions for
each source does not change substantially from the sum of NMOG and OM (Fig. S22). We then updated existing
SPECIATE profiles with volatility distributions of LVOCs and SVOCs normalized to CROC (Table S5a). Because
data on the functionality of these low volatility emissions is lacking, we assume they share similar chemical properties
(i.e. reactivity) to linear alkanes as a proxy for more complex mixtures of aliphatics and other compounds.

**2.3 Air Quality Model Configuration**

We used an updated version of the Community Multiscale Air Quality (CMAQ) model v5.3.2 to quantify the impact
of the new mobile emissions on regional-scale air quality (U.S. EPA, 2021; Appel et al., 2021). Hourly ambient air
concentrations of OA and $O_3$ were simulated for the entire year 2017 at 12 km horizontal resolution with inputs from
EPA's air QUAlity TimE Series (EQUATES) project (U.S. EPA, 2022c; Foley et al., 2023). Meteorology was
simulated with WRFv4.1.1. The Biogenic Emission Inventory System (BEIS) predicted biogenic gas emissions online
in CMAQv5.3.2. Gas- and aerosol-phase chemistry are modeled with the Carbon Bond 6 mechanism (CB6r3_AE7)
with updates for production of SOA from mobile IVOCs implemented by Lu et al. (2020) Anthropogenic emissions
are described in the US EPA 2017 emission platform technical science document and EQUATES documentation (U.S.
EPA, 2022a, c). Mobile emissions for 2017 were recalculated in order to update speciation and apply both
IVOC/NMOG and CROC/OM adjustments. The 'CMAQ-ROC' simulation implements all revisions to mobile
elemental carbon (EC) speciation described in section S2 and the methods described in sections 2.2.1 and 2.2.2. The
EC speciation updates result in substantial changes to nonroad diesel, aircraft, marine and rail source (Table S9).
Because MOVES uses source- and species-specific emission rates for HAPs rather than relying on generic speciation
of NMOG, ROC updates for HAPs are not propagated to the air quality model simulations, although we show potential
changes to national-scale HAP emissions from updates to VOC speciation. Volatile chemical product (VCP) emissions
are simulated for 2017 with the VCPy tool (Seltzer et al., 2021). Nonoxygenated and oxygenated IVOC emissions
from VCPs are represented with the IVOC chemistry from Lu et al. (2020), which results in an average SOA yield of
approximately 30% at ambient conditions across all IVOCs. However, Pennington et al. (2021) found the oxygenated
IVOC SOA yield to be 6.28%, though this yield warrants re-evaluation with better speciation and yield data given the
diverse mix of oxygenated IVOCs with varying molecule functionalities that can influence SOA production (Humes
et al., 2022). Based on available information, we reduce the CMAQ-predicted VCP SOA concentrations by 33.8% to
account for the overrepresentation of SOA from VCP oxygenated IVOCs (see section S7).
We assess model performance for $O_3$ and OC during the 2017 model year with aily-averaged measurements at routine
monitoring sites. We also perform a separate CMAQ simulation for comparison that is consistent with the EQUATES
project, which assumes the speciation of OM emissions from all sources are consistent with the volatility distribution
of a small diesel generator (Robinson et al., 2007). This 'EQUATES' simulation also utilizes the simplified potential-





combustion SOA (pcSOA) approach used in publicly available versions of CMAQ (Murphy et al., 2017). The CMAQ-
ROC simulation neglects pcSOA since the role of mobile and VCP IVOC SOA formation are explicitly accounted
for. Finally, we also analyzed two simulations with mobile and VCP SOA precursors each set to zero to quantify direct
sector contributions to total OA. This approach does not account for the contributions these sectors make to the
atmospheric oxidant capacity through emissions of low molecular weight VOCs and nitrogen oxides.
**3. Results and Discussion**
**3.1 Volatility-Resolved Mobile Source ROC Emissions**
Using the 2016 annual predictions from MOVES and the other mobile emission models processed and speciated with
the 'ROC' approach, we explore for the first time a complete bottom-up inventory of organic carbon emissions from
mobile sources in the U.S. Figure 1 shows the results of the ROC and Conventional approaches for one example
source, onroad heavy-duty diesel equipped with particulate filters. Non-organic particulate matter species such as ions
and other PM are equivalent in both approaches. Nonvolatile OM emissions in the Conventional approach are
distributed in the ROC approach to a range of SVOCs and IVOCs, which are predominantly alkanes and branched
compounds for diesel sources. The magnitude of emission factors for compounds in the VOC volatility range from
onroad diesel sources are reduced by 47.8% due to the introduction of IVOCs (IVOC/GROC = 52.2%), and the
distribution of VOC functionality is changed substantially due to adoption of VOC speciation profiles from Lu et al.
(2018). Unknown ROC mass is also reduced from 7% of total emissions to 0.7% after introducing IVOCs. Emission
factors vary by orders of magnitude across mobile sources, motivating careful accounting of sampling biases (Figs.
S18-S21), which requires the ROC approach in the emission modeling workflow to be complex and involve multiple
tools and intermediate steps (Fig. S1).
Figure 3 shows the predicted contributions of source types and functional groups across the volatility spectrum for
2016 ROC inventory. The VOC emissions are roughly evenly distributed between onroad and nonroad sources (1130
and 1045 kt yr$^{-1}$, respectively), IVOCs are weighted towards onroad (62%), and CROC (i.e. SVOCs and larger
compounds) is roughly split among onroad, nonroad, and others. Tailpipe (i.e. exhaust) emissions while running
represent the majority across all volatility categories (56% of total ROC), although evaporative sources are important
in the VOC range (38%), and similar to prior estimates (Gentner et al., 2009). It could be counter-intuitive, given
laboratory data on start and idle emission factors, that the start/idle operating mode does not contribute more to total
ROC emissions. This result could be due in part to substantially more time spent by sources in the running mode
during normal operation, but it could also be partly due to MOVES neglecting start modes for nonroad sources. Drozd
et al. (2018) found that cold start IVOC fuel-based emission factors are about 6 times larger than those from hot-
running-start emissions for newer vehicles, which is consistent with the post Tier 2 gasoline vehicles in this work. For
older vehicles though, the ROC inventory predicts greater IVOC emissions factors for hot-running modes than cold-
start for older vehicles (Table S1a and Table 2). Further research is needed to constrain NMOG emission factors and
IVOC/NMOG ratios for older (pre-2004) vehicles that are expected to have contributed approximately 72% of onroad
gasoline ROC emissions during 2017 (see Fig. S24 and Table S1a).



Emissions from gasoline-fueled sources dominate the VOC range in Fig. 3, but diesel-fueled sources, of which there
are far fewer in the U.S. dominate the IVOC range. Whereas, sources using both fuels are important for CROC
emissions. Mobile source VOCs comprise many functionalities, and aromatics make a substantial contribution. The
higher volatility IVOCs have mass associated with aromatics from gasoline sources, but cyclic hydrocarbon
compounds contribute to IVOCs across all volatilities, a feature reported by Zhao et al. (2015) We currently lack data
to specify CROC functionality across all mobile categories, so we have labeled them alkane-like based on observations
of motor vehicle POA emissions (Worton et al., 2014). Improved CROC speciation is needed, especially given the
importance of functionality to SOA formation (Lim and Ziemann, 2009; Yee et al., 2013).
**3.2 Impact of Filter Artifacts**
Transitioning from the Conventional approach to the ROC approach has implications for near-source particle
concentrations and prompt SOA production. Figure 4 shows the contributions of mobile categories with results using
approaches from previous work (Murphy et al., 2017; Lu et al., 2020). The Conventional approach assumes all OM
stays in the particle phase, which has been shown to lead to poor AQM performance (Murphy et al., 2017). The
'Robinson et al.' case, which is consistent with CMAQv5.3.2, applies the volatility distribution for a small nonroad
diesel engine, where half the OM mass is assumed to be IVOCs adsorbed to filters and is thus volatilized. As seen in
Fig. 4, only 25% of the OM persists in the particle after evaporation in the 'Robinson et al.' approach. Lu et al. (2020)
applied gasoline and diesel-specific volatility profiles parameterized for emissions from in-use vehicles to the entire
mobile category, leading to less evaporation of OM than the 'Robinson et al.' approach. Lu et al. (2020) also applied
a conversion factor of 1.4 to all mobile gasoline-fueled sources to account for missing SVOCs.
In the ROC approach here, we apply source-specific adjustment factors (Table S5) and volatility profiles (Table S6)
and find similar results for onroad gasoline and nonroad diesel compared to Lu et al. (2020). However, onroad diesel
CROC emissions are increased by 60% relative to the CROC emissions from the 'Lu et al.' approach, driven by the
inclusion of missing SVOCs from clean test conditions for diesel engines with DPFs. Conventional OM emissions
from nonroad sources are greater than those from onroad for both gasoline- and diesel-fueled sources. Nonroad
gasoline emissions reduced by 36% relative to 'Lu et al.' where emission factors are large, and CROC/OM is much
less than 1.0 (Table S5), indicating the presence of IVOCs on the filter. Predicted conventional OM emissions from
air, rail, and marine sources are also important, and CROC emissions are slightly larger than OM. Across the mobile
sector, total CROC emissions increased by 12% relative to OM, and 42% of the CROC emissions are predicted to be
in the particle phase at 298 K and 10 µg m$^{-3}$ organic aerosol (OA) loading.
**3.3 National-Scale Impact on PM, O$_3$ and HAPs**
When aggregated across all mobile sources, total ROC emissions are nearly identical between the Conventional
approach and ROC approach (Fig. 5). Total IVOC emissions are represent only 10.2% of total GROC due to the
substantial role of VOCs from gasoline sources to ROC emissions in the U.S. The spatial distribution of IVOC and
CROC emissions highlight the key role of cities, highways, and shipping lanes (Fig. S26). We calculate the OA
potential as the sum of particle-phase mass (calculated at 298 K and 10 µg m$^{-3}$) for each species and the SOA yield of
the vapor-phase component of each species. Mobile source OA potential has contributions from all ROC volatility



classes with 6.8% from LVOCs, 25.4% from SVOCs, 19.1% from IVOCs, and 48.7% from VOCs (Fig. 5). The
estimated VOC OA potential is mainly driven by adjusted yields of aromatic VOCs, which are enhanced over previous
work due to corrections for vapor wall-losses of single-ring aromatic yields (Zhang et al., 2014). These metrics
possibly reflect an upper bound on VOC and IVOC contribution as they apply SOA yields to the precursor emission
without consideration of reaction rates, timescales, or competitive losses of precursors and intermediates to deposition.
Potential OA relative contributions from air, marine, and rail (12%) and onroad diesel (16%) sources play a larger
role in OA potential when emissions are estimated with the ROC approach, while nonroad gasoline and diesel (38%)
and onroad gasoline potential OA (34%) decrease (Fig. 6). While aromatic species dominate OA potential in the VOC
precursor range, in the IVOC range OA potential has larger contributions from cyclic alkane compounds from onroad
diesel sources (Fig. S23). In the LVOC range and below, the ROC approach assumes only alkane-like species;
improvements to the SPECIATE database and emissions modeling tools will support increased detail on compound
functionality when provided by future studies.
VOCs account for 97% of the ozone potential approximated by maximum incremental reactivity (MIR), and the total
ozone potential decreases by 8.9% due to the shift in mass from VOC to IVOC. The national-scale source distribution
of $O_3$ potential changes little between the Conventional and ROC approaches (Fig. 6). Ozone potential is dominated
by onroad and nonroad gasoline sources in the highest ROC volatility bins, driven by alkane, aromatic, and oxygenated
species, as expected (Fig. S23). Among onroad light duty gasoline vehicles, 72% of ROC emissions, 68% of $O_3$
potential, and 79% of OA potential are predicted to come from pre-Tier 2 vehicles, while these vehicles account for
19% of the fuel used in 2017 (Fig. S25). Heavy-duty diesel vehicles without particulate filters or selective catalytic
reduction systems contribute 87% of ROC emissions, 85% of $O_3$ potential, and 91% of OA potential while using 31%
of the fuel for the heavy-duty diesel onroad category.
National-scale HAP emissions changed substantially with updates in VOC speciation and introduction of IVOCs with
many species decreasing by nearly 20% or more including toluene (-19%), hexane (-22%), 1,3-butadiene (-34%), and
ethyl benzene (-29%) and others increasing substantially including formaldehyde (+22%), acrolein (+20%), and
acetaldehyde (+19%) (Fig. S25). These results emphasize the need for more research on HAP emission factors, but
we keep them constant for the CMAQ simulations to focus on OA and $O_3$ changes.
**3.4 Air quality model results**
Mobile ROC emissions were generated for the year 2017 to be comparable with the EQUATES 2017 emission inputs.
Differences between the EQUATES mobile inputs and those for the CMAQ-ROC simulation (Table S9) are consistent
with the changes in the 2016 emissions results depicted in Fig. 4. The CMAQ-ROC simulation predicts lower OC
concentrations throughout the domain due to elimination of pcSOA. CMAQ-ROC predictions compared well against
both $O_3$ and OC measurements at Air Quality System (AQS) sites in 2017 (Figs. S28, S29 and Table S10). Normalized
mean biases for OC improved (in absolute terms and on average) by 11.3% in spring, 4.3% in autumn, and 7.6% in
winter. In summer, the OC underprediction increased by 12%. Overprediction in the northeast, Ohio Valley, Upper
Midwest, and northwest in winter is consistent with timing and geography of residential wood combustion emissions,
which may be overrepresented in both simulations. Root mean square error and correlation coefficient differences



between the EQUATES and CMAQ-ROC simulations are small. CMAQ predicts both the annual mean and variability
of OC concentrations well at selected U.S. cities (Fig. S34, S35), with the exception of New York City where the
model overpredicted OC by more than a factor of 2.
The predicted annual population-weighted average OA attributable to mobile sources is 0.26 µg m$^{-3}$, or 9% of the OA
from all anthropogenic and biogenic sources. Mobile source contributions to POA and SOA are similar on average,
with apparent spatial differences (Fig. 7). Average total mobile source OA appears stable between winter and summer
seasons (Fig. S30), and this is a result of trade-offs between higher POA concentrations in winter and higher SOA in
summer (Figs. S31, S32). In rural areas, model-predicted mobile OA contributions asymptote at 4.5% of total OA,
and in some urban areas they can exceed 23% (annual averages; Fig. S33). The ratio of SOA to OA is equal to 70%
in rural areas and decreases with increasing population to 20-40%. Diurnal profiles at select cities indicate SOA
formation peaks at noon in Los Angeles, Denver, Chicago and New York, but that feature is not reproduced on average
at Houston and Raleigh (Figs S34, S35).
CMAQ-ROC mobile and VCP IVOC concentrations are enhanced in urban areas with minimal seasonal differences
predicted (Figs. S36, S37). Mobile sources are predicted to contribute 20-25% to total IVOCs depending on location
and time of year, while VCP sources contribute 59-66% (Fig. S36), although IVOCs from other sources are
underrepresented. The composition of ambient IVOCs predicted by CMAQ-ROC and the speciation of IVOC
emissions from mobile and VCP emissions are consistent with results from Zhao et al. (Fig. S38). Since ambient
IVOC concentration measurements for 2017 are lacking, we extrapolated concentrations to the CalNex campaign in
2010 and find acceptable agreement with campaign-average hydrocarbon and oxygenated IVOC observations (section
S8, Fig. S39a,b). Extrapolation of CMAQ-ROC SOA to 2010 underpredicts mean CalNex SOA observations by 46%
(Fig. S41c,d). Potential explanations include underestimated emissions from other sources (e.g. cooking),
mischaracterized chemical processing (e.g. SOA yields), or errors in modeling regional pollution in Southern
California (Lu et al., 2020).
The U.S. annual GROC emission rate for mobile (2.49 Tg yr$^{-1}$) is 20% less than that of VCPs (3.09 Tg yr$^{-1}$), but the
mobile IVOC emissions (0.25 Tg yr$^{-1}$) are only one third those of VCPs (0.77 Tg yr$^{-1}$). Gas-phase oxidation is
responsible for less than half (42% and 44%) of the loss of mobile and VCP SOA-froming GROC, but 88-90% of the
IVOC loss (Fig. 8). The annual production and loss of total OA from mobile and VCPs is similar, and loss is distributed
evenly across deposition processes and transport out of the model domain. The annual rate of OA production (emission
plus chemical production) estimated by CMAQ and normalized to total ROC emissions (i.e. the sum of NMOG plus
conventional OM) is 0.16 g OA (g ROC)$^{-1}$, which is approximately equal to that estimated from the data in Fig. 5. This
agreement is surprising considering that the latter calculation does not account for variations in OA partitioning, NO$_x$
effects on SOA yields, or competitive losses from wet scavenging and dry deposition. Seasonal trends for OA, SOA
and POA production rates and ambient concentrations normalized to OM and NMOG emissions are tabulated in Table
S11 and discussed in section S9. These data may inform simple (e.g. screening) models of the impact of anthropogenic
emissions on human exposure.



### 4. Conclusions

This study implements a detailed source- and species-level procedure for converting conventional OM and NMOG mobile emissions to metrics compatible with the most recent science and speciation developed for atmospheric ROC. Although many AQMs have implemented online or pre-processing emission adjustments to account for these phenomena, (Koo et al., 2014; Murphy et al., 2017) the procedure should be embedded within emission models and databases for several reasons. Most importantly, this detailed approach considers a more diverse population of sources of different ages, fuels, and control technologies that are typically averaged together before they are passed to the AQM. Additionally, the new procedure enables near-explicit speciation of each emission source before mapping to model species used in a particular chemical mechanism. Having a detailed speciation of major emission sources is critical for assessing and revising chemical mechanisms (Pye et al., 2022b). Finally, operationalizing conversions from OM to CROC and NMOG to GROC alleviates AQM users from the burden of interrogating their emissions files to determine whether complex scaling operations are needed. From the broader perspective of facilitating transfer of knowledge between the scientific and regulatory communities, the SPECIATE database is now capable of ingesting speciation profiles with factors aligned with the most recent research studies and has enhanced flexibility to accommodate future updates. Nonetheless, for model applications seeking to scale legacy emission inputs, we provide updated factors normalized to several levels of source aggregation in Table S12 and discuss the uncertainty introduced with this approach in section S10.

The 2016 ROC emissions suggest slight decreases to total $O_3$ formation due to reapportionment of VOC to IVOC in this approach, but 2017 CMAQ-ROC predictions do not meaningfully change when evaluated at AQS sites. Meanwhile, mobile IVOC emissions enhance OA formation by an additional 79 kt yr$^{-1}$ compared to estimates from the EQUATES configuration (319 kt yr$^{-1}$). Gaps between total OA measurements and CMAQ-ROC predictions will be addressed through improved modeling of other sources of ROC (e.g. VCPs, wildfires, residential wood combustion, and cooking). Within the mobile sector, results indicate substantial contributions from onroad (46%) and nonroad (41%) gasoline and somewhat less from onroad (5%) and nonroad (3%) diesel air, marine, and rail sources (4.7%; Fig. 6). The vast majority of ROC emissions and impacts are attributable to older (pre-Tier 2 light duty gasoline and non-DPF heavy duty diesel) vehicles and nonroad gasoline engines. Onroad pollution will continue to decrease as these vehicles are phased out, increasing the importance of other mobile source categories and other sources.

This study suggests several specific uncertainties pertaining to mobile source emissions need further laboratory and field investigation. Developing complete ROC volatility distributions for specific source classes and control types is critical, especially within the nonroad category where fewer experimental data were available for this study. The CROC/OM factors are uncertain across all mobile sources. Ideally, IVOC and CROC emissions should be sampled by a filter and a broad-spectrum adsorbent tube in series to avoid filter artifacts (Khare et al., 2019). If filter-based methods alone are used to inform organic aerosol emission inventories, then reducing the uncertainty in the relationship between particle emission factor and total CROC will strengthen our confidence in estimating organic aerosol emissions, particularly for lower-emitting technologies. Some CROC/OM ratios derived for this work are between 0.85 and 1.15, indicating a limited role for partitioning bias during source testing in those cases, but many



are greater than 1.30, especially the lower-emitting sources. Lastly, more research is needed to determine the extent
to which NMOG measurements capture IVOCs (quantified by the IVOC/NMOG or IVOC/GROC ratios). These
parameters are especially important to understand for older vehicles and equipment which drive historical and
contemporary emissions. We recommend that emissions tests specifically measure and report CROC and GROC to
facilitate comparison among datasets and implementation in emission models. Currently, these measurements are
beyond the scope of typical regulatory requirements, and future progress requires research beyond regulatory methods.

## ASSOCIATED CONTENT

The Supporting Information is available free of charge at
Supporting Information 1 (SI-1): Word Document
Supporting Information 2 (SI-2): Excel Sheet with Tables
The CMAQ model source code used is available via Zenodo (https://doi.org/10.5281/zenodo.7869142). The functions
to estimate OA and $O_3$ potential are available at https://github.com/USEPA/CRACMM.

## ACKNOWLEDGMENT

The authors gratefully acknowledge contributions from U.S. EPA staff including Kristen Foley and George Pouliot
who for emissions inputs and Chad Bailey, Michael Hays, and Sergey Napelenok for internal technical reviews. We
also acknowledge Yunliang Zhao of the California Air Resources Board for valuable insights and consultation.

## AUTHOR INFORMATION

**\*Corresponding Author:** Benjamin N. Murphy; Address: Center for Environmental Measurement and Modeling,
109 TW Alexander Dr., Durham, NC 27709, USA; Email: murphy.ben@epa.gov; Phone: 919-541-2291

### Author Contributions

The manuscript was written and revised through contributions of all authors. All authors have given approval to the
final version of the manuscript. DS made contributions to the study primarily when employed by US EPA.

## DISCLAIMER

*The views expressed in this article are those of the author(s) and do not necessarily represent the views or the policies*
*of the U.S. Environmental Protection Agency*

## COMPETING INTERESTS.

*Some authors are members of the editorial board of ACP. The peer-review process was guided by an independent*
*editor, and the authors have also no other competing interests to declare.*

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



**Table 1.** Definitions of key terms.

| Acronym | Definition |
| --- | --- |
| OM | Organic matter component of primary particle emissions as measured on a filter. |
| NMOG | Non-methane organic gas emissions |
| POA | Primary organic aerosol. Particle-phase emissions after equilibrium is reached with ambient conditions. |
| OA | Particle-phase organic material at ambient conditions. |
| LVOC | Low-volatility organic compounds ($C^* \leq 0.32$ µg m$^{-3}$). |
| SVOC | Semivolatile organic compounds ($0.32 < C^* \leq 320$ µg m$^{-3}$). |
| IVOC | Intermediate volatility organic compounds ($320 < C^* \leq 3.2 \times 10^6$ µg m$^{-3}$). |
| VOC | Volatile organic compounds ($3.2 \times 10^6$ µg m$^{-3} < C^*$). |
| CROC | Condensable reactive organic carbon: particle- and gas-phase LVOC + SVOC. Carbon and noncarbon mass are included. |
| GROC | Gaseous reactive organic carbon: particle- and gas-phase IVOC + VOC. Carbon and noncarbon mass are included. |
| ROC | Reactive organic carbon – all particle and gas organic compounds mass except methane. Carbon and noncarbon mass are included. |




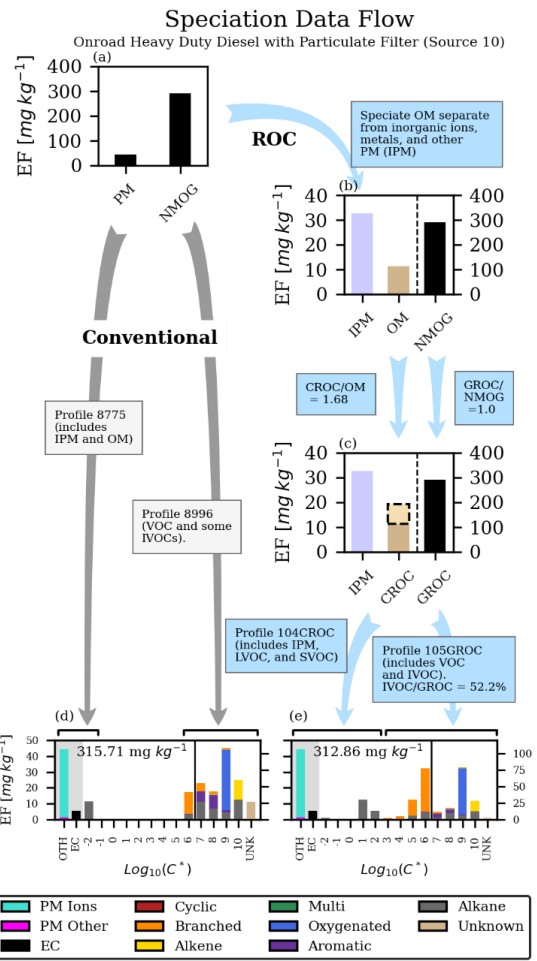


**Figure 1.** Depiction of calculation steps for the Conventional and ROC approaches to speciation of PM and NMOG emissions. Panel (a) shows the reported fuel-based emission factors based on MOVES predictions for 2016. Panel (b) shows the inorganic ions, metals and other nonorganic matter (IPM) separated from organic matter (OM). The beige area inside the dashed box in panel (c) indicates emissions that are added in the conversion of OM to CROC to account for underrepresented SVOCs from the filter measurement. Panels (d) and (e) show the comprehensive emission factors for the Conventional and ROC approaches, respectively, with data arranged by volatility while indicating non-organic PM emissions as well. In panels (d) and (e), bars to the left and right of the vertical line at $Log_{10}(C^*) = 6.5$ are quantified by the left and right y axes, respectively. The number within panels (d) and (e) indicates the total ROC emission factor excluding EC and Other PM for onroad heavy-duty diesel sources. 'Alkane' refers to only linear alkanes, while 'cyclic' and 'branched' are cyclic alkanes and branched alkanes. 'Multi'



indicates multifunctional organics. The bars in the gray shaded regions are not included in the organic volatility
distribution but are included in the CROC-compatible SPECIATE profiles (e.g. 104CROC).



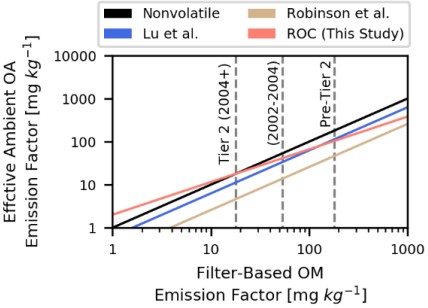


**Figure 2.** Effective ambient primary organic aerosol emission factor estimated at 298 K and 10 µg m$^{-3}$ as a function
of the OM emission factor for onroad gasoline-fueled vehicles.




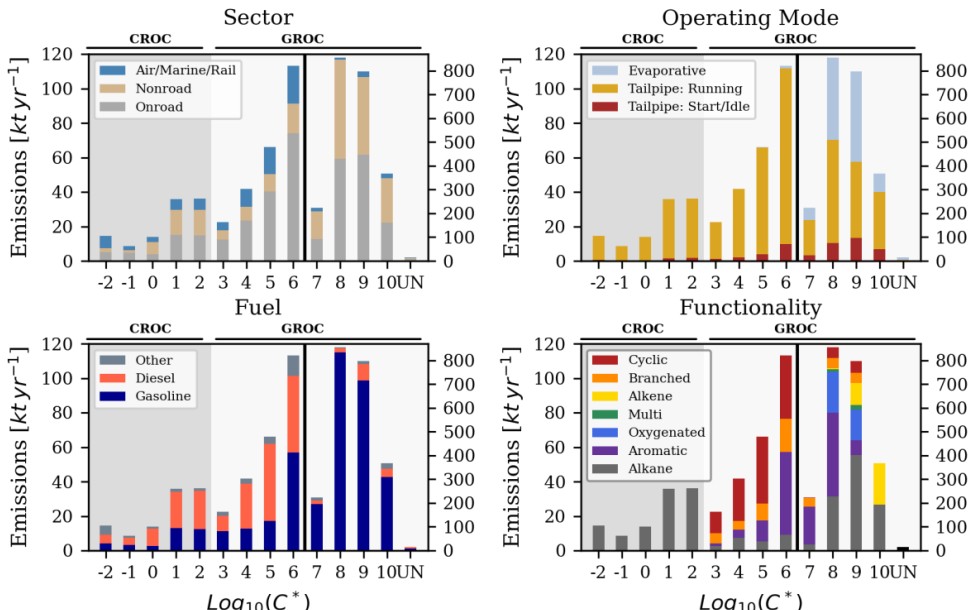

**Figure 3.** Volatility-resolved mobile source ROC emissions for the contiguous U.S. during 2016 stratified along several dimensions including category (top-left), operating mode (top-right), fuel (bottom-left), and chemical functionality (bottom-right). The 'multi' functionality series corresponds to compounds that are both oxygenated and have double carbon bonds. Bins to the left of the solid black line are quantified by the left y axis and those to the right by the right y axis. The unknown emissions (UN) are not assigned to a volatility bin and do not contribute to OA or $O_3$ formation.



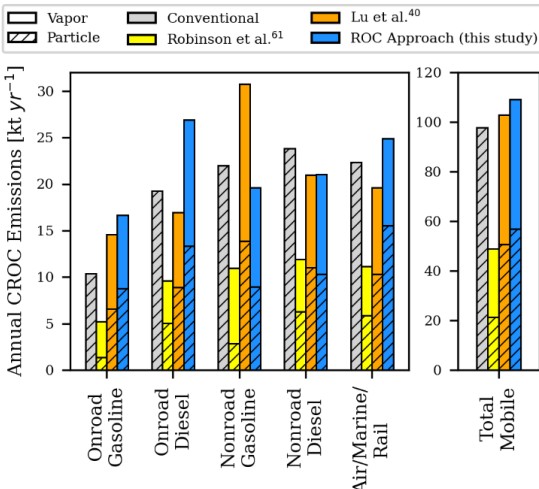

664

**Figure 4.** Bottom-up predictions of 2016 annual mobile CROC (i.e. SVOC, LVOC, and lower volatility compound) emissions classified by category, model approach, and equilibrium phase distribution. The full height of each bar corresponds to total CROC emissions. Gas-particle partitioning is calculated for atmospherically relevant conditions at 298 K and organic aerosol loading of 10 µg m$^{-3}$.




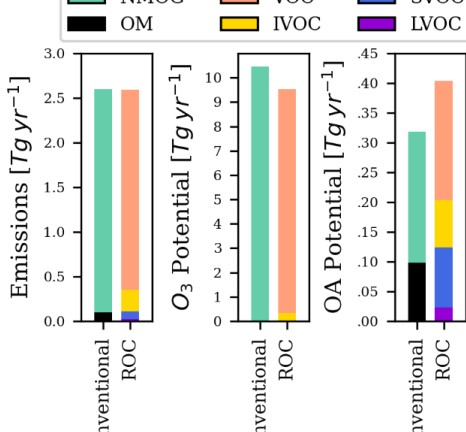


**Figure 5.** Total U. S. mobile source emissions for 2016 with aggregate O$_3$ and OA potential calculated at the species level.




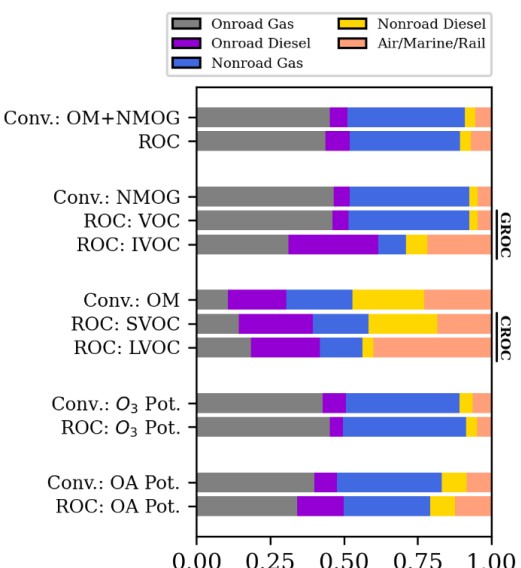

**Figure 6.** Mobile sector contributions to ROC classes and derived quantities like $O_3$ and OA potential. Values are
presented for the Conventional and ROC-based approaches.

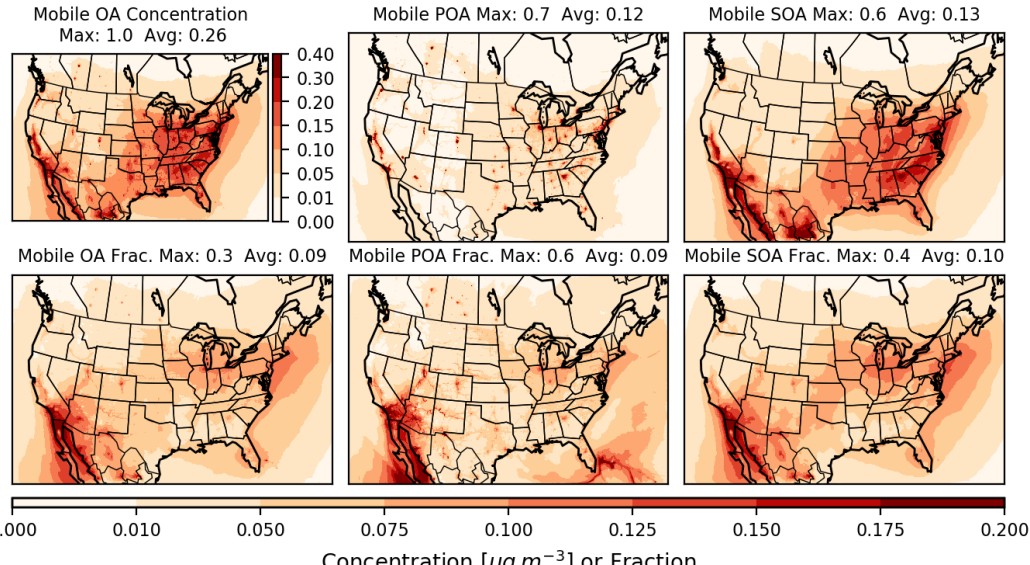


**Figure 7.** Annual average concentration (top row) of total OA (left), POA (center), and SOA (right) from mobile
sources predicted by CMAQ for 2017 with the ROC mobile emission inventory. The fractional contribution of mobile
sources to the total of each pollutant category from all sources are on the bottom row. In all panel subtitles, 'Max'
refers to the spatial maximum of the annual average spatial field, while 'Avg' refers to the population-weighted
average of the annual average spatial field.





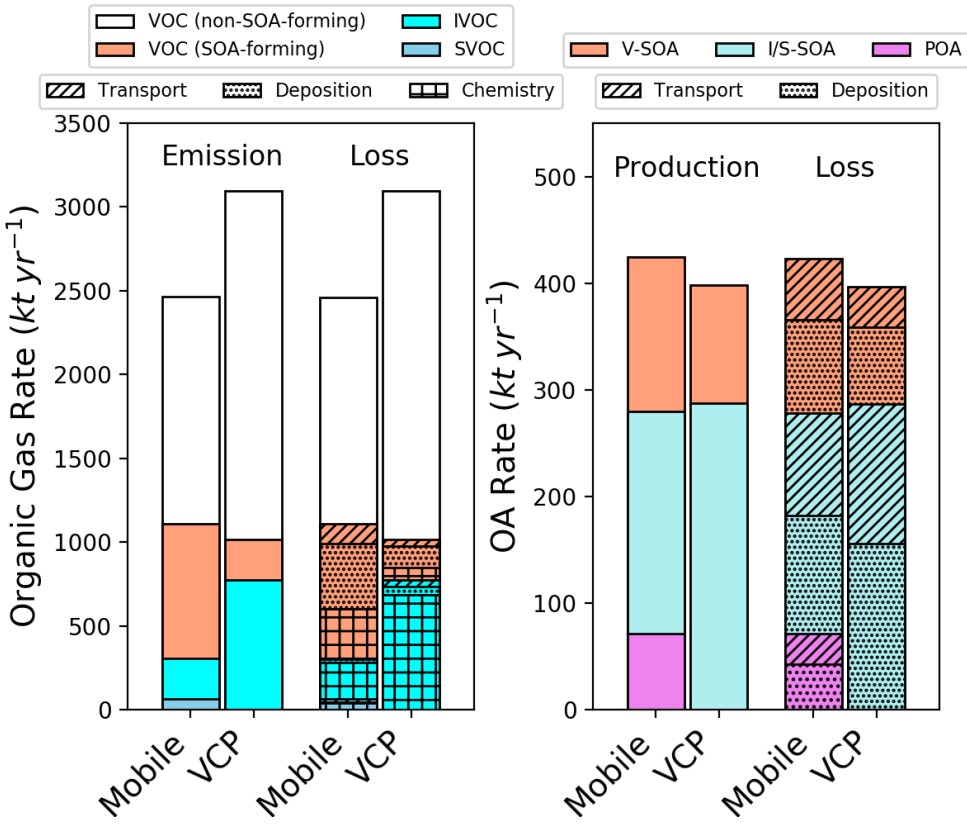

**Figure 8.** Domain-wide predicted budget of (left) mobile and volatile chemical product (VCP) gas-phase emissions and loss due to chemistry, deposition, or transport and (right) OA production and losses for 2017. In the left plot, loss terms are only depicted for categories of compounds that lead to organic particle formation.