# Peer review of "Reactive Organic Carbon Air Emissions from Mobile Sources in the United States"

_EGUsphere, 2023_

## Author Comment (AC1)

Egusphere-2023-855

Response to Reviewers

**Reviewer 1**

The authors have constructed comprehensive emission data of organic compounds from mobile sources by utilizing conventional emission data of nonmethane organic gas (NMOG) and particulate organic matter (OM). The developed emission data include information about volatility and detailed chemical composition of organic compounds, and are important basis for accurate simulation of atmospheric PM and O3. This manuscript is well written, and clearly organized.

I recommend this manuscript for publication after the following minor comments are addressed.

We thank the reviewer for helpful comments on this study. Please see responses below.

1. Line 168: I could not follow the procedure for converting OM emissions to CROC emissions. I guess that gas-particle partitioning during filter sampling is critical in this conversion and that sampling temperature and OM concentrations are key parameters. However, you calculate EF_CROC as a function of EF_OM, and you did not consider sampling temperature and OM concentrations in this conversion. Did you assume that EF_OM is proportional to OM concentrations at sources, and that effect of sampling temperature is negligible? I would like this point to be clearly stated.

   The reviewer is correct. A major finding of this study is that for sources of similar engine technology, fuel, and control technology, we find that it is possible to represent the relationship between EF_CROC and EF_OM without considering variations in temperature and OM concentration. This simplified approach is potentially limited to mobile sources because temperature is highly controlled during the execution of test method protocols (i.e. 47 degrees C). Temperature is used to calculate $c^*$ of partitioning components and then calculate total CROC (e.g. Fig. S4). Because the CROC emission factor is so tightly correlated with OM emission factor, we argue that the power law function associating them accounts for variation due to the underlying volatility distribution and the increase in concentration with emission factor. In the future, this uncertainty should be reduced by reporting and cataloging CROC emission factors rather than OM, so that temperature and concentration are not an issue. We add the following text to the methods section:

   *"These datasets show that it is possible to represent the relationship between OM emission factor and CROC emission factor without explicitly considering variations in temperature and OM concentration. This simplified approach is limited to mobile sources because temperature is tightly controlled by test method requirements (i.e. 47 ℃). Temperature is used to calculate the saturation concentration of partitioning components and then calculate total CROC (e.g. Fig. S4). Because the resulting CROC emission factor is highly correlated with OM emission factor, we argue that simplified functions*

*associating them account for variations due to the underlying volatility distribution and increases in concentration with emission factor."*

We have also added the following to section S6:

*"With the Lu et al. (2018) profiles we can estimate total CROC emissions from the organic aerosol emission factors reported by May et al. (2013b), as shown in equation S3.*

$$EF_{CROC} = EF_{OM} \cdot [CROC]/[OM] \tag{S3a}$$

$$[OM] = \left( \sum_{i=-2}^{5} \frac{f_{10^i}}{1+c_{10^i}^*(T)/C_{OA}} \right) \cdot [ROC_{\leq 10^5}] \tag{S3b}$$

$$[CROC]/[ROC_{\leq 10^5}] = \frac{f_{CROC}}{f_{\leq 10^5}} \tag{S3c}$$

$$[CROC]/[OM] = \frac{f_{CROC}}{f_{\leq 10^5}} \left( \sum_{i=-2}^{5} \frac{f_{10^i}}{1+c_{10^i}^*(T)/C_{OA}} \right)^{-1} \tag{S3d}$$

$$c_{10^i}^*(T) = c_{10^i,298\,K}^* \cdot \left( \frac{298}{T} \right) e^{\frac{-\Delta H_{vap,i}}{R} \cdot \left( \frac{1}{T} - \frac{1}{298} \right)} \tag{S3e}$$

$$\Delta H_{vap,i} = 85 - 11 \cdot log_{10} c_{10^i,298\,K}^* \tag{S3f}$$

*Equation S3a states that we can relate CROC emission factors to OM emission factors using the ratio of calculated total CROC concentration to the OM concentration measured in each test. Equation S3b accounts for partitioning from low volatility species (c\* bin centered at $10^{-2}$ μg m$^{-3}$) up to IVOCs (c\* bin centered at $10^5$ μg m$^{-3}$) and calculates the total gas plus particle mass across this entire volatility range. We select c\* = $10^5$ μg m$^{-3}$ as a nominal upper bound because higher volatility compounds are not expected to partition significantly to the filter sample and most volatility distributions reported in the literature do not exceed this IVOC bin. It adjusts this total by the ratio $\frac{f_{CROC}}{f_{\leq 10^5}}$ where $f_{\leq 10^5}$ is the total fraction of ROC mass from the c\* bin centered at $10^5$ μg m$^{-3}$ and below, while $f_{CROC}$ is the fraction of mass from c\* bin centered at 100 μg m$^{-3}$ and below, which meets the CROC definition. The c\* values used in equation S3d are adjusted for temperature (typically 47 °C) using equation S3e. Enthalpy of vaporization is parameterized in May et al. (2013a,b) as equation S3f. When Eq. S3a-f are applied to systems beyond this study, the upper bound volatility bin in equation S3d may be reduced from c\* bin centered at $10^5$ μg m$^{-3}$ to a lower volatility bin, as long as it at least includes the c\* bin centered at 100 μg m$^{-3}$ so that all CROC is captured."*

2. Line 220: daily-averaged measurements?

   Corrected.

3. Numbering of Figures and Tables should be carefully checked. (e.g., CROC/OM ratio in Line 284 is in Table S6 (not Table S5). Fig. S41 (Line 351) and Fig. S42 (Section S10) are not shown in the Supplementary Material.

Corrected.

**Reviewer 2**

This article aims at detailing the speciation of reactive organic compounds in OM and NMOG emissions. This article provides many details on the specification, which may be very useful for other modellers. However, many papers already estimate reactive organic compound specifications from OM and NMOG emissions, and the introduction does not sufficiently explain what is done in the other papers. Also, for clarity, the measurement biases that this article aims to correct should be introduced and explained in a paragraph in the introduction.

Thank you to the reviewer for thoughtful comments and helpful suggestions that improve the explanation of this study's motivation and contemporary scientific literature findings. We expanded discussion of the recent rigorous approaches to treating ROC emissions in inventories throughout the world:

> *"Manavi and Pandis (2022) and Sarica et al. (2023) implement emission factors and speciation of SVOCs and IVOCs specific for mobile sources in Europe, while Morino et al. (2022) explores revisions to stationary source ROC emissions in Japan. Chang et al. (2022) implements a more detailed bottom-up inventory of ROC emissions across all sectors in China with emission factors specified at the volatility bin level rather than for bulk PM and NMOG."*

We have added some additional explanation and references for measurement biases in the introduction:

> *"However, laboratory and field measurement campaigns have demonstrated that much of the mobile source POA is subject to gas-particle partitioning and filter sampling artifacts. These artifacts may bias the interpretation of filter-based measurements by yielding higher POA emission factors due to the presence of these adsorbed vapors (Robinson et al., 2010; Bessagnet et al., 2022)."*

Commented [MB(1):] Add Turpin et al. (https://doi.org/10.1016/1352-2310(94)00133-6)

Minor comments :

1. Line 60 : Is IVOC more impacted than SVOC for filter artefacts ? Please explain why.

   This introductory discussion refers to the fact that at high organic aerosol concentrations (e.g. greater than 100 μg m$^{-3}$), IVOCs will be expected to absorb to the filter sample. Meanwhile, at excessively low concentrations, IVOCs may bias filter measurements through adsorption to the sample. Because IVOCs are more abundant in the composition of ROC emissions than SVOCs and because IVOCs are not expected to partition significantly to the particle phase at typical ambient concentrations, they will contribute as much or more to absorptive artifacts than SVOCs. Although the impact of IVOCs on filter artifacts should be considered, the filter artifacts do not directly impact the calculation of IVOC in this study because IVOC speciation is based on NMOG emission factor measurements (converted to GROC). We have removed the clause at the end of the sentence so that it will not confuse readers.

2. In the introduction line 64 and thereafter, please detail the main assumption currently used in the litterature to estimate IVOC and SVOC. Although IVOC may be specified from NMVOC or GROC, it may also be directly speciated in the measurements (see for example Sarica et al. Env. Pollution, doi:10.1016/j.envpol.2023.121955)

We have modified the first sentence of this paragraph to explicitly point out that SVOCs and IVOCs are scaled to POA or NMOG. We have also added this interesting Sarica et al. (2023) publication to the list of references included at the end of the paragraph as another example of a study that uses updated speciation for a group of specific (in this case) mobile sources to arrive at a better representation of IVOCs and SVOCs.

However, we do not see a fundamental difference between specifying IVOC from NMOG or GROC and the approach applied in Sarica et al. (2023) as that study also speciated emissions of POA (assumed to be entirely low volatility organic compounds) and NMVOC (assumed to be VOC, IVOC, and SVOC) with updated profiles. This is generally the approach we apply as well. Chang et al. (2022), on the other hand, introduce greater detail in their emission inventory by including actual SVOC and IVOC emission factors. That level of detail would be ideal to include in future versions of the National Emission Inventory in the US and the European inventories (e.g. developed with the COPERT methodology).

3. What is the advantage to define GROC ? In the methodology defined here, GROC/NMOG ratio and IVOC/GROC ratio with speciation for each of them need to be specified. Why is it better than what is usually done, i.e. simply define a speciation for VOC and determe a IVOC/NMVOC ratio ?

The added value of converting NMOG to GROC is that NMOG is an operationally defined value based on a particular measurement of the gas-phase component of organic compounds present. Meanwhile, GROC describes the total gas plus particle concentration of compounds based on their intrinsic properties. In this way, GROC emissions can be compared across tests with varying conditions and across sources with different compounds. Differences between NMOG and GROC arise primarily from the contribution of SVOCs to NMOG (they will be excluded from GROC). Another source of uncertainty arises from limitations in the flame-ionization detection method for measuring NMOG emissions. This method often misses or underrepresents the contributions of oxygenated compounds.

This study simplifies the conversion from NMOG to GROC because there is not yet enough certainty to suggest large differences between the two metrics for mobile sources. For sources like gasoline vehicles, this may be because SVOCs are expected to contribute little to the total NMOG. But for diesel, SVOCs are expected to contribute significantly. It is uncertain how much oxygenation may bias NMOG measurements, but it is expected to be much less important than this bias would be for more oxygenated systems like wood burning emissions.

The framework we have proposed is meant to be a generic approach applicable for all ROC emission sources and also flexible enough to be inclusive of future measurement data. Research is needed to better constrain GROC/NMOG ratios for mobile sources, and if these turn out to deviate from 1.0 significantly for any system, then the infrastructure will exist to incorporate that knowledge directly.

4. Line 278 : How are estimated the source-specific adjustment factors ? Where are they detailed ? Table S5 details the volatility profiles (which are key properties for SOA modelling). Where do those come from ? What are the incertainties associated to those profiles ?

We apologize for the typo in line 278. As the first reviewer pointed out, it should have read Table S6 instead of S5. The volatility profiles are all from published literature. References have been added to Table S5. The uncertainty associated with these profiles varies for each system. Generally, the parameters predict partitioning of emissions within a factor of 2 of the measurements as shown in May et al. (2013).

References

Chang, X., Zhao, B., Zheng, H. T., Wang, S. X., Cai, S. Y., Guo, F. Q., Gui, P., Huang, G. H., Wu, D., Han, L. C., Xing, J., Man, H. Y., Hu, R. L., Liang, C. R., Xu, Q. C., Qiu, X. H., Ding, D., Liu, K. Y., Han, R., Robinson, A. L., and Donahue, N. M.: Full-volatility emission framework corrects missing and underestimated secondary organic aerosol sources, One Earth, 5, 403-412, 10.1016/j.oneear.2022.03.015, 2022.

May, A. A., Presto, A. A., Hennigan, C. J., Nguyen, N. T., Gordon, T. D., and Robinson, A. L.: Gas-particle partitioning of primary organic aerosol emissions: (1) gasoline vehicle exhaust, Atmospheric Environment, 77, 128-139, 10.1016/j.atmosenv.2013.04.060, 2013a.

Sarica, T., Sartelet, K., Roustan, Y., Kim, Y., Lugon, L., Marques, B., D'Anna, B., Chaillou, C., and Larrieu, C.: Sensitivity of pollutant concentrations in urban streets to asphalt and traffic-related emissions, Environmental Pollution, 121955, 10.1016/j.envpol.2023.121955, 2023.

---

## Editor Decision (ED1)

Suggest rewording paragraph on lines 66-78 (tracked changes version) as indicated below. It was confusing with the edits in response to reviews. Also note that this is the first time ROC is introduced, and it may need to be further defined since it is a relatively new term, as is done on lines 81-83.

Some air quality models (AQMs) have incorporated SVOCs and IVOCs by scaling these emissions to sector-wide POA or NMOG inputs either during a data pre-processing step or the AQM runtime (Murphy and Pandis, 2009; Shrivastava et al., 2011; Ahmadov et al., 2012; Bergström et 69 al., 2012; Koo et al., 2014; Woody et al., 2015; Zhao et al., 2016a; Woody et al., 2016; Jathar et al., 2017b; Murphy et al., 2017). However, these approaches rely on broad application of assumptions that may not be appropriate for specific source types since sampling artifacts will bias low-emitting and high-emitting sources differently (Robinson et al., 2010). As emissions from individual combustion sources are continually reduced in response to tightening regulations, accounting for the potential biases becomes important. Manavi and Pandis (2022) and Sarica et al. (2023) implemented emission factors and speciation of SVOCs and IVOCs specific for mobile sources in Europe, while Morino et al. (2022) explored revisions to stationary source reactive organic carbon (ROC) emissions in Japan. Chang et al. (2022) implemented a more detailed bottom-up inventory of ROC emissions across all sectors in China with emission factors specified at the volatility bin level rather than for bulk PM and NMOG. Additional bottom-up approaches are needed that revise emission factors and speciation profiles for all relevant individual source types and regions.

Minor editorial comments:
A mix of past and present tense is used in the methods section. Suggest using past, or at least being consistent.

Check line 216-typo in edit.

Figure 2 is not called in the text.

Line 301: remove "are" before represent.
* * *
**Moved down [1]:** Bottom-up approaches thus are needed that revise emission factors and speciation profiles for individual source types.

**Moved (insertion) [1]**

---

## Author Response (AR2)

**Egusphere-2023-855 Response to Editor**

**Editor comments:**

Suggest rewording paragraph on lines 66-78 (tracked changes version) as indicated below. It was confusing with the edits in response to reviews. Also note that this is the first time ROC is introduced, and it may need to be further defined since it is a relatively new term, as is done on lines 81-83.

Some air quality models (AQMs) have incorporated SVOCs and IVOCs by scaling these emissions to sector-wide POA or NMOG inputs either during a data pre-processing step or the AQM runtime (Murphy and Pandis, 2009; Shrivastava et al., 2011; Ahmadov et al., 2012; Bergström et 69 al., 2012; Koo et al., 2014; Woody et al., 2015; Zhao et al., 2016a; Woody et al., 2016; Jathar et al., 2017b; Murphy et al., 2017). However, these approaches rely on broad application of assumptions that may not be appropriate for specific source types since sampling artifacts will bias low-emitting and high-emitting sources differently (Robinson et al., 2010). As emissions from individual combustion sources are continually reduced in response to tightening regulations, accounting for the potential biases becomes important. Manavi and Pandis (2022) and Sarica et al. (2023) implemented emission factors and speciation of SVOCs and IVOCs specific for mobile sources in Europe, while Morino et al. (2022) explored revisions to stationary source reactive organic carbon (ROC) emissions in Japan. Chang et al. (2022) implemented a more detailed bottom-up inventory of ROC emissions across all sectors in China with emission factors specified at the volatility bin level rather than for bulk PM and NMOG. Additional bottom-up approaches are needed that revise emission factors and speciation profiles for all relevant individual source types and regions.

Minor editorial comments:
A mix of past and present tense is used in the methods section. Suggest using past, or at least being consistent.

Check line 216-typo in edit.

Figure 2 is not called in the text.

Line 301: remove "are" before represent.

**Moved down [1]:** Bottom-up approaches thus are needed that revise emission factors and speciation profiles for individual source types.

**Moved (insertion) [1]**

**Response:**

We thank the Editor for excellent suggestions to the paragraph above. We have rewritten it as follows and submit both the track changes and clean manuscript version.

Some air quality models (AQMs) have incorporated SVOCs and IVOCs by scaling these emissions to sector-wide POA or NMOG inputs during a data pre-processing step or the AQM runtime (Murphy and Pandis, 2009; Shrivastava et al., 2011; Ahmadov et al., 2012; Bergström et al., 2012; Koo et al., 2014; Woody et al., 2015; Zhao et al., 2016a; Woody et al., 2016; Jathar et al., 2017b; Murphy et al., 2017). However, these approaches rely on broad application of assumptions that may not be appropriate for specific source types since sampling artifacts will bias low-emitting and high-emitting sources differently (Robinson et al., 2010). As emissions from individual combustion sources are continually reduced in response to tightening regulations, accounting for these potential biases becomes important. Manavi and Pandis (2022) and Sarica et al. (2023) implemented emission factors and speciation of SVOCs and IVOCs specific for mobile sources in Europe, while Morino et al. (2022) explored revisions to stationary source organic emissions in Japan. Chang et al. (2022)

implemented a more detailed bottom-up inventory of organic emissions across all sectors in China with emission factors specified at the volatility bin level rather than for bulk PM and NMOG. Additional bottom-up approaches are needed that revise emission factors and speciation profiles for all relevant individual source types and regions.

We have revised the use of tense in the Methods section and converted verbs to past tense, except where it was appropriate to use present (e.g., when explaining a known theory or principle).

Corrected line 216.

We did find several places where Figure 2 was referenced in the text, but as Fig. 2. We are willing to make additional revisions if we are misunderstanding this issue.

Removed 'are' in line 301.